# Obstructive Sleep Apnoea and Lipid Metabolism: The Summary of Evidence and Future Perspectives in the Pathophysiology of OSA-Associated Dyslipidaemia

**DOI:** 10.3390/biomedicines10112754

**Published:** 2022-10-29

**Authors:** Martina Meszaros, Andras Bikov

**Affiliations:** 1Department of Pulmonology and Sleep Disorders Centre, University Hospital Zurich, 8091 Zurich, Switzerland; 2Department of Pulmonology, Semmelweis University, 1083 Budapest, Hungary; 3North West Lung Centre, Wythenshawe Hospital, Manchester University NHS Foundation Trust, Manchester M23 9LT, UK; 4Division of Infection, Immunity and Respiratory Medicine, University of Manchester, Manchester M13 9MT, UK

**Keywords:** obstructive sleep apnoea, OSA, dyslipidaemia, lipid, metabolic dysfunction

## Abstract

Obstructive sleep apnoea (OSA) is associated with cardiovascular and metabolic comorbidities, including hypertension, dyslipidaemia, insulin resistance and atherosclerosis. Strong evidence suggests that OSA is associated with an altered lipid profile including elevated levels of triglyceride-rich lipoproteins and decreased levels of high-density lipoprotein (HDL). Intermittent hypoxia; sleep fragmentation; and consequential surges in the sympathetic activity, enhanced oxidative stress and systemic inflammation are the postulated mechanisms leading to metabolic alterations in OSA. Although the exact mechanisms of OSA-associated dyslipidaemia have not been fully elucidated, three main points have been found to be impaired: activated lipolysis in the adipose tissue, decreased lipid clearance from the circulation and accelerated de novo lipid synthesis. This is further complicated by the oxidisation of atherogenic lipoproteins, adipose tissue dysfunction, hormonal changes, and the reduced function of HDL particles in OSA. In this comprehensive review, we summarise and critically evaluate the current evidence about the possible mechanisms involved in OSA-associated dyslipidaemia.

## 1. Introduction

Obstructive sleep apnoea (OSA) is common disorder which is characterised by recurrent collapses of the upper airways during sleep. Intermittent hypoxia (IH) and sleep fragmentation are the most important factors in the pathomechanism of OSA resulting in sympathetic overdrive, oxidative stress and systemic inflammation. These derangements lead to cardiovascular and metabolic alterations, such as atherosclerosis, hypertension, insulin resistance and dyslipidaemia, ultimately contributing to cardiovascular morbidity and mortality [1].

Dyslipidaemia is an independent risk factor for cardiovascular morbidity [2]. There is also strong evidence supporting the role of OSA with altered lipid profile: elevated triglyceride (TG), total cholesterol (TC) and low-density lipoprotein cholesterol (LDL-C) concentrations with a corresponding reduction in high-density lipoprotein cholesterol (HDL-C) levels were commonly found in patients with OSA [3,4]. Understanding the mechanisms linking OSA to lipid abnormalities is of major clinical importance, as they could represent treatable traits (i.e., choosing the right lipid-lowering medication and lifestyle changes), and also highlights the importance of active screening of dyslipidaemia in patients with OSA.

The aim of this review is to summarise and critically evaluate the current evidence about the possible mechanisms involved in OSA-associated dyslipidaemia. Naturally, we focus on human studies; however, we briefly discuss animal models and highlight if the research was conducted in humans or animals.

## 2. Overview of Physiological Lipid Metabolism

### 2.1. The Physiological Role of Chylomicrons

Dietary TGs are hydrolysed by several lipases (for example pancreatic and gastric lipases) to FFAs and monoacylglycerol (MAG) to be absorbed by the enterocytes in the small intestine [5]. FFAs can be transported by passive diffusion or fatty acid transporters, such as cluster determinant 36 (CD36) or fatty acid-transport protein 4 (FATP4). Dietary cholesterol esters (CEs) are hydrolysed to FFAs and free cholesterol (FC). Several FC transporters were identified on the enterocytes, such as the Niemann–Pick C1-like protein (NPC1L1), ATP-binding cassette protein G5 (ABCG5) and G8 (ABCG8) and scavenger receptor class B type I (SR-BI). They are also expressed in the apical membrane of hepatocytes [6].

The chylomicrons (CMs) are large TG-rich lipoproteins containing apolipoprotein B-48 (apoB-48) which are usually formed by dietary FFA absorbed from the small intestine [7]. The most common role of CMs is to transport dietary cholesterol and TGs to the peripheral tissues and to the liver; the process is called “exogenous lipid transport” (Figure 1). CMs have also an active role in enterohepatic cholesterol transport. Around 1000 g of biliary cholesterol is secreted to the intestine every day. Thus, the majority of CM-transported cholesterol derives from the reabsorption of biliary cholesterol [8].

CMs are synthetised in the endoplasmic reticulum (ER) and Golgi apparatus of the enterocytes. First, in the ER, the previously hydrolysed FFAs, FC and MAG are resynthesised. FATP4 converts FFAs to fatty acyl-CoA (FFA-CoA) [9], and thus FFA-CoA and MAG can be converted to diacylglycerol (DAG) by monoacylglycerol acyl transferase 2 (MGAT2) [10,11]. Then, diacylglycerol acyl transferase 1 (DGAT1) further converts DAG with FFA-CoA to triacylglycerol (TAG) [12], which is the main component of the nascent CM [10]. TAG can leave the ER to form cytosolic lipid droplets or chylomicrons. FC is transformed to CEs by the acyl-coenzyme A:cholesterol acyltransferase (ACAT).

Apolipoprotein B-48 (apoB-48) is a unique marker of CMs, and each CM particle contains one apoB-48 molecule [13]. ApoB-48 is a truncated form of the hepatic apolipoprotein B-100 (apoB-100), and it has 48% of the initial length of apoB-100 after a post-transcriptional mRNA modification in the intestine. The scaffolding of apoB-48 with TAG, CE and phospholipids is mediated by microsomal triglyceride transfer protein (MTP), resulting in primordial CMs. Primordial CMs can be expanded with further TG or CM by MTP or after the fusion with lipid droplets [14]. After the core expansion, apolipoprotein A-IV (apoA-IV) is also assembled into the nascent CM surface [15]. Nascent CMs are transported to the Golgi in pre-CM transport vesicles (PCTVs) [16,17] where apolipoprotein A-I (apoA-I) is also incorporated. Finally, a pre-CM leaves the enterocyte through its basolateral membrane by exocytosis.

In the circulation, a nascent CM gains further apolipoproteins, such as apoC-I, apoC-II, apoC-III and apoE from HDL [18], resulting in a mature CM. After hydrolysis of TAGs from CMs by lipoprotein lipase (LpL), the core of CMs is decreased. The remnant CM particles are taken up mainly by LDL receptors (LDLRs) of hepatocytes [19].

### 2.2. The Physiological Role of VLDL

Very-low-density lipoprotein (VLDL) delivers lipids from the liver to the peripheral tissues; this process is called “endogenous lipid transport” (Figure 1). VLDL is synthetised in the liver and regulated by the FFA influx. The FFAs originate from adipocytes, CM remnants and the intestine via the portal vein [20]. VLDL is composed of a TG-rich core (synthetised from FFAs) surrounded by FC, CE, phospholipids (PLs) and apolipoproteins (apoB-100, C, E).

ApoB-100 expressed by the liver is the essential component of VLDL, IDL and LDL particles. ApoB-100 is lipidated in the ER by MTP, resulting in primordial VLDL and then pre-VLDL particles [21]. Under inadequate TG availability, apoB-100 can be degraded in several ways (such as ER-associated degradation or post-ER presecretory proteolysis). Primordial VLDL can also fuse with microsomal lipid droplets associated with apoC-III [22]. Pre-VLDLs are transported to the Golgi in specialised vesicles (VLDL transport vesicles). During the Golgi-associated maturation of VLDL, apoB-100 undergoes conformational changes and VLDL is expanded by lipoproteins [23], and VLDL leaves the Golgi by exocytosis.

VLDL is metabolised by LpL to produce intermediate-density lipoproteins (IDLs) which can be taken up by the liver via apoE receptors [24]. IDLs can transfer apoE to HDL particles, avoiding hepatic clearance, and their TG content is hydrolysed by hepatic lipase (HL), resulting in CE-rich LDL particles. VLDL receptors (VLDLRs) can be found in several tissues such as adipose tissue, muscle and heart and recognise apoB-100 and apoE. They can bind VLDL, IDL or CM but not LDL particles [25].

### 2.3. The LDL Metabolism

LDL particles are the most important cholesterol carriers in the circulation (Figure 1). LDL consists of mainly CE, FC, TG, PL and a single molecule of apoB-100 [26]. The size, composition and density of HDL are mainly influenced by LpL and cholesteryl ester transfer protein (CETP) functions [27]. Several modified LDL particles were identified, such as oxidised LDL (oxLDL), small dense LDL (sdLDL) and desialylated LDL, which are strongly atherogenic [28,29,30]. Further subclasses can be identified by gel electrophoresis: large (LDL 1–2) and small (LDL 3–7) subfractions [31]. Circulating LDL particles are absorbed by the liver (70%) and peripheral tissues (30%), mainly by the LDL receptor (LDLR) [32].

LDLR binds apoB-100 and apoE with high affinity and is thus responsible for the uptake of VLDL, IDL and LDL particles [6]. The expression of LDLR is regulated via a negative feedback mechanism mediated by the complex of sterol-regulated transcription protein-2 (SREBP-2) and SREBP cleavage activating protein (SCAP) [33]. In cholesterol-depleted cells, the SREBP-2/SCAP complex is proteolytically cleaved in the Golgi, resulting in SREBP-2. SREBP-2 activates HMG-CoA reductase and LDLR, leading to cholesterol uptake [34,35]. On the contrary, high cellular cholesterol levels lead to conformational changes of SCAP in the ER not allowing the transport of the SREBP-2/SCAP complex to the Golgi [36]. LDLR can be regulated at the posttranscriptional level by proprotein convertase subtilisin/kexin type 9 (PCSK9) which degrades LDLR [37].

Other receptors can also eliminate LDL from the circulation. Low-density lipoprotein receptor-related protein-1 (LRP-1; also known as cluster of differentiation 91 (CD91)) is a multifunctional receptor expressed in several tissues (hepatocytes, adipocytes, muscle cells, macrophages and endothelial cells). Its expression is strongly regulated by metabolic and inflammatory processes [38]. LRP-1 receptors mediate the clearance of apoE-containing lipoproteins (VLDL and CM remnants) mainly in the absence of LDLR. In the lack of hepatic LRP-1, CM clearance is decreased [39]. Moreover, it also binds lipases, such as HL and LpL: they enhance the binding of apoE-containing lipoproteins to LRP [40,41]. LRP-1 plays a role in the HDL metabolism by enhancing the recycled apoE accumulation in early endosomes [42]. The presence of macrophages with LRP-1 deletion was associated with elevated plasma CE and TG levels resulting in the accumulation of circulating TG-rich lipoproteins [43]. Other studies demonstrated the atherogenic effects of LRP-1 as it can also mediate the accumulation of cholesterol in macrophages [43] or in cardiomyocytes [44]. Besides the lipid metabolism, LRP-1 internalises more than 100 ligands, including proteinases and proteinase-inhibitor complexes (tissue plasminogen activator (tPA), urokinase PA (uPA), matrix metalloproteases (MMPs)), coagulation factors, growth factors and matrix proteins (fibronectin, thrombospondin). LRP-1 is able to regulate several transcription factors (such as nuclear factor-κB (NF-κB)) affecting immune responses and tissue survival. In pathophysiological circumstances, LRP-1 can be shed by proteases, resulting in its soluble form (sLRP-1). During inflammation, sLRP-1 stimulates the expression of further pro-inflammatory cytokines, such as TNF-α [38]. However, other studies reported its anti-inflammatory function. sLRP1 mediates the internalisation of αMβ2 integrin, resulting in the inhibition of its adhesion properties [45]. In addition, sLRP-1 decreased the expression of TNF-α and IL-1 [46].

### 2.4. The Role of Lipoprotein Lipase

LpL hydrolyses the VLDL- and CM-associated TAGs to FFAs and MAGs which are taken up by the target cells. This enzyme is mainly produced by adipose tissue and skeletal and cardiac muscle and transported to the luminal surface of endothelial cells by the glycosylphosphatidylinositol-anchored high-density lipoprotein-binding protein 1 (GPIHBP1) [47]. This endothelial LpL pool is referred to as the functional LpL [48]. LpL activity is regulated by different physiological stimuli in a tissue-specific manner. In white adipose tissue, LpL activity is increased by the postprandial state and decreased by fasting [49]. On the contrary, fasting activates the myocardial LpL [50]. Finally, in skeletal muscle, LpL activity is promoted by acute exercise [51]. ApoC-II is the main cofactor of LpL activity [18], whereas apoC-I and apoC-III have been shown to inhibit LpL activity [52]. Moreover, some members of the family of angiopoietin-like proteins, such as ANGPTL3 (hepatocyte), ANGPTL4 (adipocyte) and ANGPTL8, also promote the inhibition of LpL [53,54]. Several hormones, such as insulin, glucocorticoids and adrenalin, stimulate LpL activity in the adipose tissue [6].

### 2.5. The Physiological Role of HDL

HDL is a major mediator in reverse cholesterol transport (RCT). RCT is termed as a cholesterol transport from the peripheral cells (including macrophages) back to the hepatocytes for further metabolism [55]. In general, HDL particles comprise a hydrophobic core with CE and TG covered by PL, FC and apolipoproteins (apoA-I, A-II, A-IV, A-V, C-I, C-II, C-III, E, F, J, M). Various HDL particles highly differ in their size, shape, proportion of proteins and lipids and biological activities [56]. The two main forms of HDL are the small and poorly lipidated discoid HDL (also known as preβ-HDL) and the larger and CE/TG-containing spherical HDL (also known as α-HDL) [56,57]. Spherical particles represent the majority of HDL particles in the circulation [57]. HDL_2_ particles are larger and lipid-rich but less dense, and HDL_3_ particles are smaller, lipid-poor and dense [58]. These can further be divided into HDL3c, HDL3b, HDL3a, HDL2a and HDL2b fractions [57]. Further subclasses can be identified by gel electrophoresis: large (HDL 1–3), intermediate (HDL 4–7) and small (HDL 8–10) subfractions [31]. The small HDL 8-10 particles are atherogenic through easy penetration to the endothelium and low recognition by HDL receptors [59,60].

The main structural apolipoproteins of HDL are apoA-I (70%) and apoA-II (20%) [61]. ApoA-I plays role in activating LCAT and also has anti-inflammatory and antioxidant effects [62,63]. ApoA-II is an important inhibitor of LpL directly and indirectly by replacing apoC-II in VLDL. Moreover, it also has a cofactor activity for LCAT and CETP [64]. ApoM accounts for approximately 5% of HDL proteins. It plays role in lipid transfer into nascent HDL [65] and enhances the cholesterol efflux from foam cells [66]. Noteworthily, apoM is a carrier of sphingosine-1-phosphate (S1P) mentioned below [67]. Other apolipoproteins constitute a minor amount of HDL, such as apoA-IV, V, C-I, C-II, C-III, D, E, J and L [68]. It is important to note that apoJ or clusterin has anti-apoptotic, anti-atherogenic and anti-inflammatory properties and is involved in lipid transport forming HDL particles [69].

ApoA-I is mainly produced by the liver (70%) [70] and partly by the intestine (30%) [71]. Lipid-poor apoA-I binds ATP-binding cassette transporter A1 (ABCA1) on peripheral cells (such as hepatocytes and macrophages [72]), resulting in FC and PL transport from the cells to apoA-I [73]. Two apoA-I molecules with FC and PL form a *discoidal HDL* formation [57]. Noteworthily, these particles can also be produced from surface components of the catabolism of TG-rich lipoproteins after the LpL hydrolysis [56]. The *discoidal HDL* formation reacts quickly with lecithin cholesterol acyltransferase (LCAT) which transports free acid from lecithin to FC, resulting in CE. After esterification and incorporation of more apoA-I by LCAT, the HDL particle becomes a mature spherical form (small HDL_3_, large HDL_2_) [55,57] which is dynamically modified in the RCT. Phospholipid transfer protein (PLTP) transfers more PL and FC from VLDL to HDL, enhancing the LCAT reaction and resulting in HDL_2_ with increased size [74]. PLTP can lead to the fusion of HDL particles with a consequential production of small lipid-poor apoA-I/PL complexes [75].

The mature HDL particles can be cleared from the circulation by two main pathways: (1) The main receptor in the RCT is SR-BI, which is expressed on hepatocytes and steroidogenic cells. SR-BI has an affinity for CE and apoA-I content in HDL particles [76,77]. The hepatic HDL uptake is stimulated by HL [78]. (2) The other mechanism is the indirect pathway in which spherical HDL particles are modified by CETP. CETP is mainly produced by hepatocytes and adipocytes and circulates with HDL [79]. CETP transports CE from HDL towards apoB-containing lipoproteins (mainly LDL, but also VLDL and CM) in exchange for TG in the opposite direction. The transfer activity of CETP is regulated by the triglyceride levels [80]: in the physiological state, predominantly CEs are transported from HDL to apoB-containing lipoproteins with a minor transfer of TG in the opposite direction. In hypertriglyceridaemia, increased concentrations of apoB-containing lipoproteins are available as potential acceptors for CEs. Moreover, CETP also transports TG from TG-rich lipoproteins (VLDL, CM) to LDL and HDL, resulting in small, dense and TG-rich particles [80]. The TG and PL content of these HDL_2_ particles can be further hydrolysed by HL, resulting in lipid-poor small HDL_3_ particles which interact with ABCA1 for the next HDL circle [56,78]. The CE content of the apoB-containing particles is taken up by the hepatic LDLR.

It is important to mention that by taking cholesterol from foam cells, HDL has a protective role against atherosclerosis [81,82,83]. HDL also inhibits LDL oxidation. Small HDL_3_ particles are more resistant to oxidative damage than HDL_2_ particles and inactivate the products of LDL lipid peroxidation [84,85]. Several HDL-associated apolipoproteins [56] and HDL-bound paraoxonase-1 (PON-1) possess antioxidant properties [86]. HDL displays anti-inflammatory effects by decreasing the expression of inflammatory cytokines and adhesion molecules and inhibiting inflammatory cell activation [87,88].

## 3. Current Knowledge of The Pathophysiological Lipid Metabolism in OSA

### 3.1. Animal Models

Animal models allow experimental investigation of OSA-related processes in isolation and have been extensively used to explore the relationship between dyslipidaemia and OSA. The effects of IH were the most widely investigated [89]. These models allow researchers to precisely define the major parameters of IH, such as the frequency or the severity of the hypoxic events [90]. However, it is important to consider that experimental IH episodes cause hypoxia in animals that is significantly more severe than that experienced in humans. For a realistic stimulation of IH, SaO_2_ should be much lower in mice than the SaO_2_ observed in patients [90]. Most of the experimental studies investigated whether IH regulates the expression of different transcription factors involved in the lipid metabolism. The regulation of hypoxia-inducible factor-1 (HIF-1), SREBP-1 and stearoyl-coenzyme A desaturase 1 (SCD-1) was investigated in detail in rat and mouse models [91,92,93]. The consequences of dyslipidaemia, such as atherosclerotic lesions associated with IH, can also be studied more precisely in animal models [94].

In humans, IH and sleep fragmentation are closely interrelated [95], and animal models could better separate these entities. On the other hand, dyslipidaemia in humans is complicated by genetic factors, diet, exercise, abdominal obesity, the presence of comorbidities and medications. Therefore, complex animal models which study numerous heterogeneous processes simultaneously are warranted [94].

### 3.2. Calorie Intake in OSA

Excessive calorie intake is a main cause of aberrant obesity, which is the most important risk factor for OSA. Indeed, patients with OSA tend to consume high-calorie diets [96]. Hunger and food intake are controlled by the balance of a number of hormones, such as leptin, ghrelin, insulin, cholecystokinin, glucagon-like peptide 1 (GLP-1) and peptide YY [97]. However, increased levels of GLP-1 and gastric inhibitory polypeptide/glucose-dependent insulinotropic polypeptide were found in patients with OSA [98]. Moreover, IH seemed to upregulate the expression of peptide YY, GLP-1 and neurotensin in enteroendocrine cells [99]. These hormones have an anorexigenic influence on the enteric nervous system. As a vicious circle, sleep fragmentation in OSA attenuates hypothalamic leptin receptors, resulting in cravings for high-energy foods [100]. The consequences of this leptin resistance are an increase in fat mass and gaining weight, worsening obesity [100]. The ingested fat is the main drive of CM production [101] leading to further alterations in the lipid profile.

### 3.3. Intestinal Lipid Absorption in OSA

OSA is associated with postprandial hyperlipidaemia [102]. Indeed, using oral retinyl-palmitate, the retinyl-esters incorporated into CM had an earlier peak under IH than under normoxia [103]. Although patients with OSA have higher postprandial TG levels, experimental IH in this group did not result in a further increase in TG levels [104].

High postprandial TG levels could be due to accelerated intestinal absorption. For instance, the FFA transporter CD36 is upregulated by HIF-1 [105]. However, CD36 expression is also upregulated by the peroxisome proliferator-activated receptor-gamma (PPAR-γ), the expression of which was reported to be decreased in OSA [106]. Nevertheless, intrahepatic CD36 was increased in mice exposed to IH [107].

Bile acids act as natural detergents: they emulsify lipid dietary fat into smaller lipid droplets, making the digestion by lipases easier. Cytochrome P450 7A1 (CYP7A1), an important enzyme in bile acid synthesis, was repressed by HIF-1α under hypoxia, suggesting altered bile acid production [108]. However, the effects of IH on bile acid synthesis and absorption have not been investigated. Similarly, gastric and pancreatic lipases were not studied in OSA.

### 3.4. Impaired Intravascular Lipolysis and Uptake by the Periphery: Lpl Dysfunction in OSA

A well-described mechanism for OSA-associated hyperlipidaemia is the impaired clearance of circulating lipoproteins by LpL (Figure 2). Drager et al. showed that the functional clearance rate of CEs and TGs was significantly lower among patients with OSA compared to controls [103]. This delayed clearance was correlated with the depth of nocturnal hypoxaemia (MinSatO_2_) and disease severity (apnoea–hypopnoea index (AHI)) [109]. In human preadipocytes exposed to 24 hours of hypoxia in vitro, a 6-fold decrease in LpL activity was detected [110]. Serum LpL concentrations were lower in patients with OSA compared to controls and negatively correlated with disease severity [111].

Several OSA-associated mechanisms can lead to the altered function of LpL, including IH, oxidative stress, inflammation, catecholamines and hormones. IH itself is a potent inhibitor of LpL [103], and the degree of hypoxia correlates with the delay in TG clearance [112,113]. Serum LpL concentrations correlated with markers of nocturnal hypoxia, such as the oxygen desaturation index (ODI) [114] and nocturnal SpO_2_ [111]. In animal models of OSA, CIH increased the levels of adipose ANGPTL4 in an HIF-1α-dependent manner [103], and ANGPTL4 levels correlated with the severity of nocturnal desaturation [115]. Moreover, the antibody against ANGPTL4 increased the activity of LpL in the adipose tissue and the lung [112]. However, Mahat et al. failed to demonstrate any differences in postprandial LpL activity or ANGPTL4 expression between normoxia and IH [110], suggesting other, ANGPTL-4-independent, regulatory mechanisms during CIH [112]. In vivo, higher concentrations of plasma ANGPTL4 and ANGPTL8 were measured in patients with OSA compared to the controls [116]. Higher serum levels of ANGPTL3 were detected in patients with OSA and coronary artery disease (CAD) compared to the patients having OSA alone [117].

PPAR-γ is a main regulator of several genes associated with lipid metabolism, including LpL [118], and it is downregulated by hypoxia in an HIF-1α-dependent manner [119]. Jun et al. detected that acute hypoxia decreased the PPAR-γ expression, resulting in downregulated LpL in mice [113]. However, hypoxia had no effect on the expression of GPIHBP1, which is the carrier of LpL [113].

Inflammation was also found to impair the function of LpL in several ways. Interleukin-1 (IL-1) and tumour necrosis factor-α (TNF-α) decrease the activity of LpL in vitro [120,121] and in vivo [122], at transcriptional [123] and post-transcriptional levels [124]. Circulating LpL levels inversely correlated with CRP levels, emphasising the inhibitory role of inflammation in LpL function [111].

OSA is characterised by increased sympathetic activity [125]. Early studies indicate that catecholamines reduce LpL activity directly [126,127] and indirectly through the activation of ANGPLT4 [128].

Insulin activates LpL in the adipose tissue [129] and downregulates the expression ANGPTL3 [130]. However, insulin resistance (IR) decreases the activity of LpL. In line with this, HOMA-IR, the marker of IR, negatively correlated with LpL [111]. Leptin decreases the activity of LpL directly [131] and indirectly by decreasing the expression of ANGPTL3 [132]. Leptin levels were elevated in OSA [133]. Decreased levels of adiponectin, detected in OSA [134], were associated with lower LPL activity independently of systemic inflammation [135].

In conclusion, impaired function of LpL in OSA leads to decreased lipid uptake of the peripheral tissues resulting in an increase in circulating CM and VLDL-C levels.

### 3.5. Alternative Ways Leading to Decreased Lipid Uptake

In OSA, LRP-1 can be downregulated by SREBP-1 [136] in an FFA- [137] or IH-dependent fashion [138]. Moreover, vitamin D [139] and klotho [140], which both increase LRP-1 expression, are decreased in OSA [141,142]. The shedding of LRP-1 is facilitated by pro-inflammatory cytokines [38] or atherogenic lipoproteins [143], resulting in its soluble form (sLRP-1) which can be measured in the circulation; sLRP-1 was decreased in OSA and correlated with disease severity [144].

### 3.6. Increased Lipid Production in the Liver

The lipid production in the liver is influenced by three main mechanisms: (1) de novo lipogenesis of the hepatocytes, (2) FFA delivery and uptake from the periphery and (3) availability of lipids and carbohydrates.

Previous evidence suggested that IH activates SREPB-1, the key transcriptional factor involved in lipid biosynthesis, through HIF-1α activation [91,92]. SREBP-1 upregulates SCD-1. SCD-1 is responsible for the synthesis of monosaturated FAs (MUFAs) [93], which are substrates for PL, TG and CE synthesis [145]. As mentioned above, the HIF-1α/SREBP-1/SCD-1 pathway was widely investigated in OSA (Figure 2). Mice with partial HIF-1α-deficiency exhibited lower hepatic mRNA and protein levels of SCAP and SCD, lower hepatic protein levels of SREBP-1 and lower hepatic fat accumulation compared to the wild-type mice [92]. In a SCAP-deficient mouse model, 5 days of IH did not influence the levels of serum and hepatic lipids and expression of SREBP-1, SCD-1 and HMG-CoA-reductase [138]. Furthermore, SCD-1 deficiency in mice abolished the IH-induced increased hepatic SCD-1 and plasma VLDL-C levels and atherosclerosis in the ascending aorta [146].

The duration of IH seems to influence lipid production in OSA. Five days of IH exposure increased the serum levels of total cholesterol, HDL-C, PL, TG, hepatic TG and SREBP-1 and the protein and mRNA levels of SCD-1 [91]. However, the genetically obese leptin-deficient rats that had higher baseline lipid values did not show changes in serum lipid profile after 5 days of IH compared to the lean rats. The authors concluded that short-term IH upregulates lipid biosynthesis but does not affect it in the presence of pre-existing lipid alterations [91]. On the contrary, genetically obese rats exposed to 12 weeks of IH experienced elevated TG and PL levels as well as SREBP-1 and SCD-1 transcription [147].

The severity of IH may also affect lipid production. The ubiquitination of HIF-1α leads to the proteasomal degradation of HIF-1α protein and depends on the O_2_ tension [148,149]. In the study of Li et al., only severe IH (oxygen nadir of 5% compared to 10%) increased the hepatic SCD-1 levels [3]. The authors hypothesised that moderate IH did not prevent HIF-1α from proteasomal degradation [3]. In addition, oxidative stress contributes to hepatic lipid overproduction in two ways. Firstly, reactive oxygen species (ROS) stabilise HIF-1α [150]. Secondly, ROS induce lipid peroxidation in the liver [3]. Lipid peroxidation leads to hepatic inflammation and fibrosis resulting in nonalcoholic steatohepatitis (NASH) [151]. The pathomechanism of NASH in OSA was reviewed in detail previously by Mesarwi et al. [151].

IH also enhances hepatic lipid production through the increased sympathetic tone which has a stimulatory effect on VLDL secretion [152].

However, IH alone did not seem to be enough to cause dyslipidaemia in animal models. In atherosclerosis-resistant mice (C57BL/6J), atherosclerosis was observed only in those exposed to both IH and cholesterol-rich diet, but not in those exposed to cholesterol-rich diet or to IH alone [94]. Moreover, the combination of IH and a cholesterol-rich diet was associated with a marked progression of dyslipidaemia. The authors suggested that the presence of dyslipidaemia due to genetic or environmental factors is required for atherogenic consequences of CIH [94].

In line with this, twin studies showed genetic susceptibility to the development of dyslipidaemia [153] and OSA too [154]. In our previous twin study, we detected a heritable relationship between TG levels and sleep parameters (AHI, ODI, TST90%), suggesting a common genetic background [155]. The genetic link between OSA and TG levels has recently been confirmed in a genome-wide association study [156]. Most notably, dyslipidaemia and OSA share common genetic loci, such as PPAR-γ [157,158] or APOE polymorphism [159].

The hepatic lipid accumulation and hepatic insulin resistance can enhance the lipid alterations in OSA. The hepatic lipid accumulation is the consequence of the FFA overload from the periphery due to adipose tissue dysfunction with increased lipolysis and altered lipid clearance by LpL. The coexistence of insulin resistance may also increase VLDL production. In insulin resistance, insulin loses the ability to promote the degradation of apoB [160]. The accumulated lipid content undergoes lipid peroxidation under IH leading to NASH [151]. Moreover, the lipid overproduction leads to increased VLDL production and export to the circulation.

### 3.7. Abnormal Modifications of LDL in OSA

LDL modification is one of the most important consequences of oxidative stress and inflammation. LDL can be modified in the extracellular space or in the lysosome of macrophages [161] by enzymatic (such as myeloperoxidase (MPO)) and non-enzymatic (such as desialylation, glycosylation, interaction directly with ROS) mechanisms. Not only the lipids but also the protein components of LDL can be modified [162]. Small dense LDL (sdLDL) particles associated with hypertriglyceridaemia are often desialylated, which is the most frequent modification of LDL. Due to their decreased affinity for LDL-R, their longer circulation time makes them susceptible to other modifications [163], including glycosylation [164] and oxidation [165]. Oxidised LDL (oxLDL) particles were found to have pro-inflammatory and atherogenic potential contributing to atherosclerosis (Figure 3). OxLDL particles can be hydrolysed by PON-1 associated with HDL [166].

Pro-atherogenic sdLDL3–7 subfractions were significantly higher in the OSA group [31]. SdLDL particles were independently associated with OSA in non-obese participants [167]. LDL size was independently associated with metabolic syndrome in OSA [168]. However, Liu et al. did not detect a correlation between OSA severity measures and sdLDL [169].

Only a few studies investigated oxLDL in OSA; oxLDL levels were found to be increased in OSA in most [170,171,172,173] but not all studies [174,175]. A recent meta-analysis concluded that oxLDL levels are increased in OSA [176]. However, studies that matched in age or BMI between patients with OSA and controls showed no significant difference in oxLDL levels [176]. Furthermore, endothelial lectin-like oxidised low-density lipoprotein receptor-1 (LOX-1) was upregulated in OSA [172]. LOX-1 is the main receptor for oxLDL on endothelial cells and orchestrates the expression of adhesion molecules and may induce atherosclerosis in OSA [177].

### 3.8. HDL Dysfunction in OSA

HDL is converted to a dysfunctional form with impaired physiological effects due to IH, oxidative stress and inflammation [178] (Figure 3). The dysfunctional HDL comprises lower CE, oxidised PL, increased TG and decreased apoA-I content, serum amyloid A (SAA) and several inflammatory proteins, such as complement C3 [178,179].

There is some evidence that IH and inflammation [180] downregulate molecules in the RCT, such as ABCA1 [181] and SR-BI [91]. Short-term IH (5 days) decreased liver SR-BI protein levels independent of obesity in a mouse model. However, obese mice had lower baseline SR-B1 levels than lean mice [91]. On the contrary, long-term IH (4 weeks) did not cause a change in hepatic SR-B1 levels [3].

Oxidative stress enzymes associated with OSA [182], such as MPO, excessively oxidise HDL. The oxidative modification of apoA-I leads to its inability to interact with ABCA-1, resulting in decreased premature HDL and impaired cholesterol efflux [183,184]. Other oxidised components of HDL, such as oxidised PLs [185] or FFAs [186], can also impair the functions of apoA-I by destroying its structure [187]. Although the functionality of apoA-I seems to be altered, its levels were not affected in OSA [171]. Decreased activity of PON-1 is also associated with HDL dysfunction [188]. Circulating levels of PON-1 were lower in subjects with OSA than in controls [189,190,191,192,193].

Modified apoA-I is also not able to activate LCAT, leading to impaired RCT [194]. Moreover, oxidised HDL, through activating the NF-κB pathway [195], increases the expression of pro-inflammatory molecules, such as the adhesion molecule vascular cell adhesion molecule-1 (VCAM-1) [177]. Circulating SAA, the levels of which were elevated in OSA [196], dislocates apoA-I from HDL [197]. This SAA-rich HDL is unable to interact with ABCA-1 [198]. High calorie intake also attenuates the anti-inflammatory functions of HDL [199].

The higher levels of apoJ or clusterin in OSA [200,201] may suggest its protective function in the HDL metabolism.

Several studies evaluated the circulating HDL-C concentrations in OSA and reported decreased HDL-C levels in most [202,203] but not all cases [31]. In the study of Tan et al., OSA-associated HDL dysfunction was measured as reduced LDL oxidation by HDL [171]. Patients with OSA presented a higher degree of HDL dysfunction with a consequential higher concentration of oxLDL independent of cardiovascular comorbidities. HDL dysfunction was more strongly correlated with disease severity than HDL-C concentration [171]. In another study, HDL_2_ and HDL_3_ levels were correlated with IR, but not with OSA severity or the degree of hypoxia. The authors concluded that IR plays a role in OSA-related dyslipidaemia [169]. In a recent study, despite similar HDL-C levels between the OSA and control groups, the participants with OSA had higher pro-atherogenic small HDL 8-10 subfractions and decreased anti-atherogenic large HDL 1-3 subfractions [31]. Moreover, not only OSA severity but also sleep fragmentation was inversely correlated with HDL-C and HDL 1-3 subfractions [31].

The atherogenic index of plasma (AIP) is a biomarker of atherosclerosis and coronary heart disease which is calculated as log(TG/HDL-C) [204] and reflects the dysregulation between anti- and pro-atherogenic lipoproteins. Previous studies found significantly higher AIP values among participants with OSA compared to the controls [205,206,207,208,209]. AIP was higher in patients with OSA and associated with disease severity [206,207,209] and daytime sleepiness in some [209] but not all studies [208].

### 3.9. Increased Intracellular Lipolysis in Adipose Tissue

Fatty acids are mainly stored in the form of TAG in adipocytes [210]. This storage can be mobilised in three main steps: *(1)* Adipocyte triglyceride lipase (ATGL) catalyses the hydrolysis of TAG to DAG and FFAs [211]. *(2)* The hydrolysis of DAG is catalysed by hormone-sensitive lipase (HSL), resulting in MAG and FFAs [212]. *(3)* Finally, monoacylglycerol lipase (MGL) completes the hydrolysis, producing FFAs and glycerol [213].

Dysregulated peripheral lipolysis has been associated with OSA (Figure 2). IH leads to increased sympathetic activity [214], and elevated levels of catecholamines are major activators of lipolysis [215]. In healthy subjects, increased sympathetic tone with consequential higher HSL expression was detected after two weeks of IH [216]. In mice, IH-induced lipolysis and decreased adipocyte size were detected [217]. In line with this, IH resulted in an increase in lipolysis rate by 211% and a decrease in intracellular lipid stores by 37% in human adipocytes too [218]. However, IH did not seem to affect postprandial lipolysis in lean healthy men [110].

Oxidative stress stimulates both HSL [219] and ATGL [220]. Moreover, several lipolysis-stimulating cytokines, such as TNF-α [221] and IL-6 [222], are detected in increased concentrations in OSA [223,224].

Endothelin-1 (ET-1) is upregulated by IH and induces lipolysis through the phosphorylation of HSL [225]. Fatty acid binding protein-4 (FABP-4) facilitates lipolysis by binding HSL [226], and its levels were detected in elevated concentration in OSA [227,228,229]. FABP-4 also interacts with a co-activator of ATGL, enhancing TAG hydrolysis [230].

Obesity is associated with higher basal levels of lipolysis [231]. Leptin exerts lipolytic activity [232], whilst adiponectin has an inhibitory effect on catecholamine-induced lipolysis [233]. In line with this, increased levels of leptin [234] and decreased levels of adiponectin [235] were reported in OSA.

Insulin is the main negative regulator of lipolysis. Insulin resistance is associated with the loss of the suppressive effects of insulin [236]. Moreover, the anti-lipolytic effect of insulin depends on the O_2_ tension of adipose tissue [237]; in hypoxia, it seems to be inhibited [238].

It is important to note that fragmented sleep leads to the nocturnal secretion of adrenocorticotropin and cortisol [239], which enhance lipolysis [240].

## 4. Further Mechanisms in OSA-Associated Dyslipidaemia

### 4.1. Adipose Tissue Dysfunction

Obesity is the most important risk factor for OSA. At least 30% of obese patients have OSA, and 60% of the patients with OSA are obese [241,242]. The dysfunction of adipose tissue is an important contributor to the metabolic consequences of OSA [243]. White adipose tissue (WAT) is the most important energy storage. High levels of circulating FFAs force WAT to store lipids via two mechanisms: through increases in the number (hyperplasia) and the size (hypertrophy) of the adipocytes [244]. In contrast to hyperplasia, hypertrophy induces pathological changes in the adipose tissue by activating stress pathways, such as endoplasmic reticulum stress, oxidative stress and inflammation [245]. IH induces specific changes in WAT even in the absence of obesity [246]. However, adipocyte hypertrophy and hyperplasia are not always present in IH-induced adipose tissue dysfunction. Some previous studies detected shrunken adipocytes in the WAT of non-obese mice exposed to IH [247,248]. Moreover, IH reduced fat mass by inducing lipolysis [217]. Whereas the morphological changes of WAT are different between IH and obesity, they share the consequential abnormalities.

#### 4.1.1. Inflammation in Adipose Tissue

The larger size of adipocytes reduces the vascularity of hypertrophic adipose tissue, resulting in lower oxygen tension and hypoxic damage. The consequential hypoxia contributes to inappropriate angiogenesis mediated by vascular endothelial growth factor (VEGF) [249]. Furthermore, IH activates HIF-1α and NF-κB, consequently resulting in an increased production of cytokines and adipokines [243].

In contrast to the healthy state characterised by anti-inflammatory immune cells, such as M2 type macrophages, T-helper 2 (Th2) cells, regulatory T cells and anti-inflammatory mediators (IL-10 or adiponectin), hypertrophic WAT is infiltrated by pro-inflammatory immune cells, mainly by CD8+ cytotoxic T cells and Th1 cells leading to the production of pro-inflammatory cytokines (TNF-α, IL-6) [246]. Moreover, hypoxic and inflammatory changes result in macrophage polarisation from M2 type to M1 type. In lean mice exposed to IH, reduced M2-type and increased M1-type macrophage infiltration were also detected in adipocytes [250]. M1-type macrophages enhance the inflammation, producing further cytokines, such as monocyte chemoattractant protein-1 (MCP-1). MCP-1 is an important regulator of macrophage tissue infiltration and chemotaxis of monocytes [251]. Moreover, it is secreted from adipose tissue to the circulation and may increase the hepatic expression of SREBP-1 [251]. Increased plasma levels of MCP-1 were detected in patients with OSA irrespective of obesity and correlated with ODI [252,253]. Furthermore, in the presence of IH, human adipocytes have a higher sensitivity to express pro-inflammatory genes [254].

#### 4.1.2. Role of Adipokines

Leptin is a master regulator of food intake and body energy balance, and its levels were shown to be increased in obesity [255], diabetes [256] and cardiovascular diseases [257,258]. Leptin levels were widely investigated in OSA and found to be increased [133,259,260,261,262,263] even after adjustment for obesity [261]. OSA-associated hyperleptinaemia was related to disease severity measures, such as AHI [133,235,259,260], TST90% [235] and MinSatO_2_ [261,264]. However, high levels of leptin contribute to leptin resistance by downregulating its cellular responses [265]. Leptin resistance with the loss of physiological functions of leptin also plays a role in OSA-associated metabolic alterations [266]. In a recent animal model, leptin injection did not decrease the food intake of rats exposed to IH [267]. Moreover, IH resulted in a reduced expression of leptin receptors, suggesting the role of leptin resistance in OSA [267,268]. Sleep fragmentation attenuates leptin signalling in the hypothalamus, resulting in consequential high-calorie food intake enhancing obesity [100]. However, sleep fragmentation itself was not found to influence circulating leptin levels [269]. Obese patients with OSA have dysfunctional adipose tissue with adipocyte hyperplasia which increases leptin production [270]. Independently of obesity, IH can itself induce leptin secretion via activating the sympathetic nervous system, renin–angiotensin system and hypothalamic–pituitary–adrenal axis [246,266]. Moreover, leptin gene expression is induced by HIF-1α [271].

Leptin may contribute to lipid alterations in OSA. Leptin activates hepatic lipid production [152] and peripheral lipolysis [232] through the activation of the sympathetic nervous system and by increasing the expression of SREBP-1 and SCD-1 [272]. Moreover, it decreases the activity of LpL [131]. The dissociation between high leptin levels and its action is caused by leptin resistance and attenuated leptin signalling in the liver [273]. A recent study found that leptin levels in OSA correlated positively with TG and negatively with HDL-C concentrations [274]. Leptin can lead to oxidative stress by activating the nicotinamide adenine dinucleotide phosphate (NADPH) oxidase [275].

Adiponectin is another important adipokine with anti-inflammatory and antioxidant properties, and its levels are inversely correlated with various disorders, such as obesity [276] and hypertension [277]. Lower adiponectin levels were detected in patients with OSA compared to controls [134,278] and were correlated with disease severity independently of obesity [279]. However, some studies found comparable [280] or even higher [274] adiponectin levels in patients compared to controls. IH suppresses adiponectin expression directly and indirectly by increased sympathetic activation [281]. Adiponectin increases the production of apoA-I and ABCA1 and induces HDL assembly [282,283]. It positively correlates with HDL-C levels independent of obesity [284]. Adiponectin enhances the catabolism of VLDL by activating LpL [285]. Moreover, it increases the mRNA levels of the VLDL-R in skeletal muscle cells [286]. In line with this, there is a negative correlation between VLDL-C and adiponectin levels [287].

Another anti-inflammatory and antioxidant adipokine is omentin, the levels of which were detected in lower concentrations and correlated positively with HDL levels in OSA [280].

### 4.2. Altered Hormone Production

Several other hormones have an impact on the lipid metabolism, such as cortisol [288], growth hormone (GH) [289] and insulin [290]. GH deficiency is known to be associated with lipid alterations [291], and GH levels were decreased in OSA [292]. Cortisol overproduction is strongly associated with dyslipidaemia [293], and its levels were detected in high concentrations in OSA [294]. Insulin activates LpL in the adipose tissue [129] and inhibits lipolysis [236]. Moreover, it promotes the degradation of apoB [160], leading to decreased hepatic production of apoB-containing lipoproteins. As OSA is associated with insulin resistance, these effects are mitigated.

### 4.3. Sleep Stages

It is known that rapid eye movement (REM) sleep is associated with higher sympathetic tone [295]. REM and non-REM (NREM) sleep influence the production of several hormones, such as cortisol [296] and GH [297]. GH is mainly produced during N3 sleep [297]. Some patients have a disproportionally higher burden of obstructive events in REM than in non-REM sleep. These patients have a higher risk for hypertension, diabetes and cardiovascular disease [298].

Only a few studies investigated the association between sleep stages and OSA. Interestingly, AHI measured in the REM phase (AHI_REM_) correlated with TG levels only in one study [299], and it did not have any correlation with lipid parameters in another study [300]. Xu et al. found an independent association between AHI_REM_ and increasing levels of TG, HDL-C and apoE. However, this association became insignificant after analysing only the patients who had an AHI_NREM_ or AHI_REM_ < 5/h [301]. In contrast, AHI_NREM_ correlated with TG, apoB [299,301], HDL-C, apoA-I [299], LDL-C and cholesterol levels [301]. Slow wave sleep duration and REM latency were independently and inversely associated with cholesterol and LDL-C levels [302]. In conclusion, it could be postulated that NREM sleep may have the greatest impact on lipid alterations in OSA.

## 5. Direct Consequences of Dyslipidaemia

### 5.1. Endothelial Dysfunction

Endothelial dysfunction is defined as an impairment in the vasodilatory ability of the vessels (mainly due to the compromised nitric oxide (NO) availability) leading to altered oxygenation, oxidative stress, vascular inflammation and consequential atherosclerosis. IH has a direct detrimental effect on endothelial function [303,304,305,306,307]. OxLDL particles also impair eNOS function by decreasing its expression [308], decreasing L-arginine availability [309]. Moreover, oxLDL increases iNOS expression and ROS generation [308].

### 5.2. Systemic Inflammation and Consequential Atherosclerosis

OxLDL particles increase the levels of adhesion molecules (VCAM-1, P-selectin) on the endothelium, resulting in enhanced leukocyte recruitment [310]. In OSA, these molecules are also overexpressed by IH and oxidative stress in an NF-κB-dependent fashion [311,312,313]. This leads to increased adhesion between leukocytes and endothelium cells, resulting in the adhesion of circulating leukocytes to the endothelium and slowing down the rolling of leukocytes, thus facilitating their extravasation [314]. Moreover, the oxLDLs have a greater affinity for scavenger receptors, such as LOX-1 on endothelial and smooth muscle cells [315] and CD36 on macrophages [316]. Thus, the activated macrophages increase their CD36 expression, facilitating uncontrolled oxLDL uptake [317], and release pro-inflammatory cytokines (IL-1, TNF-α) [318]. This activation of innate immunity is a key mechanism in foam cell formation in atherosclerosis. It is important to know that adaptive immune cells, such as B-cell-derived plasma cells, are also activated and produce antibodies against oxLDL, and antigen-specific T cells produce further cytokines, resulting in enhanced inflammation [319].

HDL dysfunction in OSA also contributes to atherosclerosis [87,88]. The anti-inflammatory and anti-atherogenic effects of HDL are mainly mediated by sphingosine-1-phosphate (S1P). S1P decreases the expression of several inflammatory cytokines (such as TNF-α) and increases the expression of eNOS [320], improving endothelial function [321]. Elevated S1P enrichment was found in HDL_3_ particles [322]. HDL particles also enhance the eNOS function by binding to SR-BI expressed on endothelial cells [323]. HDL is an important inhibitor of platelet activation and aggregation as well as of coagulation factors, such as factor X and tissue factor [324].

### 5.3. Insulin Resistance

Dyslipidaemia can cause insulin resistance. Increased FFA levels due to increased lipolysis reduce insulin-mediated glucose uptake in skeletal muscle by interrupting insulin signalling [325]. Moreover, FFAs activate the NF-κB pathway, resulting in the production of pro-inflammatory cytokines such as TNF-α, IL1β and IL6 in the peripheral tissues. Systemic low-grade inflammation reduces the responsiveness of the peripheral tissues to insulin, leading to insulin resistance [326].

## 6. The Effect of OSA Therapy on the Lipid Metabolism

### 6.1. The Effect of CPAP Therapy

Continuous positive airway pressure (CPAP) is the gold standard treatment for OSA [327]. Several studies investigated the effect of CPAP on plasma or serum lipid profile in OSA. Various duration of CPAP (i.e., from 8 weeks to 6 months) effectively decreased TG, TC, LDL-C and apoB and increased HDL-C levels [328,329,330,331,332]. However, these effects depended on sufficient therapy adherence in some cases [331]. On the contrary, some studies failed to demonstrate improvement in lipid levels; the TG, TC and HDL-C levels did not change after 6 weeks to 4 months of CPAP therapy [333,334,335,336].

The effect of CPAP therapy on the lipid profile was also investigated in meta-analyses. Nadeem et al. evaluated 29 articles including 1958 participants with therapy durations ranging from 2 days to 1 year [337]. They concluded that there was a significant reduction in TC (−5.66 mmol/L) and LDL-C (−0.49 mmol/L) levels; however, TG levels did not change (−0.05 mmol/L). HDL-C levels increased after the therapy (+0.21 mmol/L) [337]. Xu et al. analysed the results of six studies including 456 subjects with therapy durations of 2–24 weeks [338]. CPAP therapy sufficiently reduced only the TC levels (−0.15 mmol/L). TG (0.00 mmol/L), LDL-C (−0.04 mmol/L) and HDL-C (−0.02 mmol/L) levels were not different between CPAP and the sham CPAP/control groups [338]. According to their subgroup analysis, younger subjects, more obese patients and patients with a longer duration of CPAP showed a significant decrease in TC concentrations (−0.27, −0.24 and −0.20 mmol/L). The authors postulated that CPAP therapy may not have any clinical effect on circulating lipid levels [338]. In the meta-analysis of *Lin* et al., six studies with 699 subjects met the inclusion criteria [339]. The time of the therapy was 4-24 weeks. Significant improvements in TC (−6.23 mg/dL), TG (−12.60 mg/dL) and HDL-C (−1.05 mg/dL) levels were detected but LDL-C concentrations did not decrease (−1.01 mg/dL) after CPAP therapy. Moreover, moderate-to-severe OSA, daytime sleepiness, CPAP treatment with short-term duration and good compliance were associated with the changes in lipid profile [339]. In a recent paper by Chen et al., 14 studies with 1792 subjects were included [340]. The therapy duration was 4-48 weeks. The CPAP therapy significantly decreased the TC levels (−0.09 mmol/L); however, it failed to change the levels of TG (0.07 mmol/L), LDL-C (−0.06 mmol/L) or HDL-C (−0.03 mmol/L). The authors did not find any confounders of CPAP treatment effect on lipid profile changes [340].

CPAP may improve some aspects of dyslipidaemia. For example, CPAP decreases the levels of several inflammatory molecules by mitigating hypoxia [341], reduces sympathetic activity [342], decreases the levels of cortisol [343] and improves insulin sensitivity [344]. CPAP increased the LpL concentrations after 3–6 months in patients with OSA [111,114]. The fractional clearance rate (FCR) of TG showed a 5-fold increase after 3-month CPAP therapy, but the FCR of CE was unchanged [109]. Circulating FFAs, which are the markers of increased lipolysis, were decreased after CPAP [345]. In line with this, CPAP withdrawal dynamically increased nocturnal FFA levels [346]. CPAP reduced the markers of lipid peroxidation, such as malondialdehyde levels [347], and decreased the endothelial LOX-1 expression [348]. However, it did not influence the oxLDL levels after 1 year of therapy in patients with OSA having comorbidities [170].

In summary, the previous studies investigating the effect of CPAP on lipid profiles were inconclusive. The studies were heterogeneous with different designs and sample sizes. The negative results of some studies may suggest that CPAP treatment alone does not improve lipid profiles in patients with OSA. Dyslipidaemia in OSA is strongly associated with comorbidities, such as obesity, insulin resistance and cardiovascular diseases, which also need to be addressed with pharmacological interventions. Furthermore, the differences between CPAP trials could be due to differences in diet, which was often uncontrolled in these studies. Most importantly, the effect of CPAP on triglyceride levels was more pronounced and more sustainable when it was combined with weight loss [349].

### 6.2. The Effect of MAD Therapy

A mandibular advancement device (MAD) is an alternative therapy option for OSA [350]. Only a few studies evaluated the impact of MAD on lipid profile in OSA. Interestingly, Recoquillon et al. detected a significant increase in TG levels after 2 months of effective MAD therapy, whilst the other investigated lipid parameters (TC, LDL-C, HDL-C) were unchanged [351]. There was no improvement in lipid profile after 12 months of MAD therapy in the study of Venema et al. [352]. Silva et al. compared the effectiveness of MAD on the metabolic profile with CPAP: CPAP was more effective in reducing TC and LDL-C levels compared to MAD therapy after 12 months [353].

### 6.3. The Effect of Upper Airway Surgery

The effect of upper airway surgery on the lipid profile in OSA has been poorly investigated. Li et al. investigated the postoperative lipid profile in patients with OSA who underwent uvulopalatopharyngoplasty (UPPP) or nasal surgery [354]. In patients who underwent UPPP, serum TC and HDL-C levels were significantly improved. In patients who underwent nasal surgery, these values did not change. Patients with isolated hypertriglyceridaemia showed significant improvements in serum TG and HDL-C levels [354]. Another study detected a UPPP-induced decrease in TG and TC levels after a 3-year follow-up [355].

## 7. Discussion of Major Findings and Further Research Directions

As outlined above, intermittent hypoxia, oxidative stress and consequential systemic inflammation may result in lipid alterations in OSA. Although most of the studies investigating these pathways were performed in vitro or in animal models, the results were also confirmed in humans. Although large population-based studies are concordant in OSA-related dyslipidaemia, they usually did not control for diet, regular exercise or lipid-lowering medications, which could contribute to bias. Clinical studies on large groups of patients are warranted to control for these factors. Furthermore, multiple mediators that are involved in dyslipidaemia (see Section 2) have not been investigated in OSA yet.

Coexistent disorders, such as obesity, insulin resistance and nonalcoholic steatohepatitis, may also lead to systemic inflammation and dyslipidaemia. This could be a reason for inconclusive results with CPAP on lipid profile. CPAP treatment alone may not be able to improve the lipid profiles in patients with OSA. Thus, parallel treatment of these comorbidities is essential to improve dyslipidaemia. Studies should also focus on which patients benefit the most from an intervention with CPAP.

As dyslipidaemia is strongly linked to OSA, patients should actively be screened for lipid abnormalities and cardiovascular complications. The detailed lipid profile of the patients with OSA should be measured at the screening visit and later under the CPAP therapy. Patients with lipid abnormalities detected during OSA management should be also referred to the appropriate specialty. Compared to single lipid components, the use of lipid components in combination with measures of abdominal obesity could better select those patients who are at higher cardiovascular risk [356].

## 8. Conclusions

In summary, OSA is associated with altered lipid metabolism and results in elevated circulating lipid levels. Intermittent hypoxia, oxidative stress and inflammatory mechanisms lead to altered lipid profiles in OSA. Dyslipidaemia promotes endothelial dysfunction and consequential atherosclerosis leading to increased cardiovascular morbidity and mortality. However, OSA-associated comorbidities might enhance these alterations. Further well-designed studies investigating potential causative associations between dyslipidaemia and OSA and involving CPAP treatment are warranted. The studies in the future should also take into consideration the role of OSA-related comorbidities in the pathomechanism of OSA-related dyslipidaemia. We strongly advocate measuring blood lipids in patients with OSA to estimate and ultimately reduce cardiovascular risk in clinical practice.

## Figures and Tables

**Figure 1 biomedicines-10-02754-f001:**
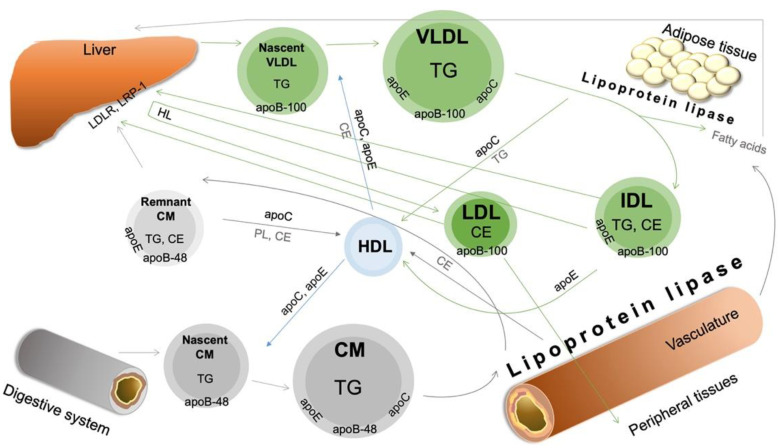
Overview of physiological lipoprotein metabolism. Apo—apolipoprotein; CE—cholesteryl ester; CM—chylomicron; HDL—high-density lipoprotein; HL—hepatic lipase; IDL—intermediate-density lipoprotein; LDLR—low-density lipoprotein receptor; LRP-1—LDL receptor-related protein 1; PL—phospholipid; TG—triglyceride; VLDL—very-low-density lipoprotein. For description and references, please see the text.

**Figure 2 biomedicines-10-02754-f002:**
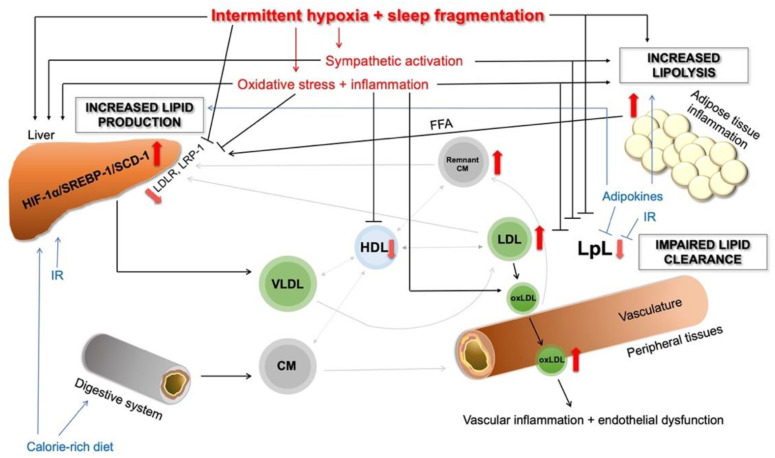
Overview of the main pathophysiological pathways in OSA-related dyslipidaemia. The red arrows show the altered pathways and dysregulations. CM—chylomicron; FFA—free fatty acid; HDL—high-density lipoprotein; HIF-1α—hypoxia-inducible factor 1 alpha; IR—insulin resistance; LDL—low-density lipoprotein; LDLR—low-density lipoprotein receptor; LpL—lipoprotein lipase; LRP-1—LDL receptor-related protein 1; oxLDL—oxidised-LDL; SCD-1—stearoyl-coenzyme A desaturase 1; SREBP-1—sterol regulatory element-binding protein 1; TG—triglyceride; VLDL—very-low-density lipoprotein. For description and references, please see the text.

**Figure 3 biomedicines-10-02754-f003:**
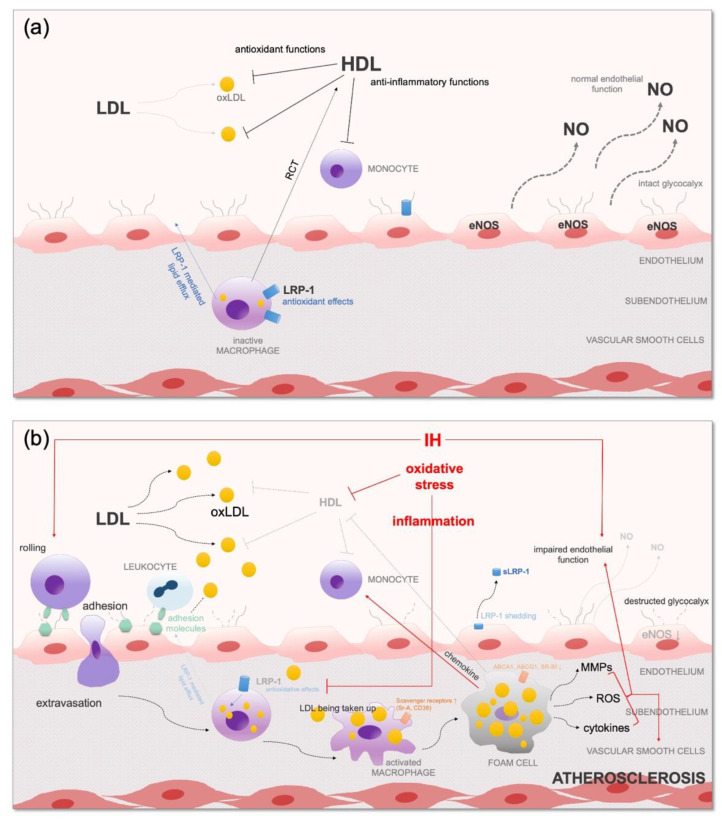
(**a**) Antioxidant effects of HDL and LRP-1 and normal endothelial function; (**b**) impaired antioxidant mechanisms with consequential oxidative stress, inflammation and endothelial dysfunction in OSA. ABCA1—ATP-binding cassette transporter A1; ABCG1—ATP-binding cassette transporter G1; CD36—cluster determinant 36; CM—chylomicron; eNOS—endothelial nitric oxide synthase; HDL—high-density lipoprotein; IH—intermittent hypoxia; IL—interleukin; LDL—low-density lipoprotein; MMP—matrix metalloproteinase; NO—nitric oxide; oxLDL—oxidised-LDL; RCT—reverse cholesterol transport; ROS—reactive oxygen species; sLRP-1—soluble LDL receptor-related protein 1; Sr-A—macrophage scavenger receptor; SR-BI—scavenger receptor class B type I; VLDL—very-low-density lipoprotein. For description and references, please see the text.

## Data Availability

Not applicable.

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
