# Peer review of "Obstructive Sleep Apnoea and Lipid Metabolism: The Summary of Evidence and Future Perspectives in the Pathophysiology of OSA-Associated Dyslipidaemia"

_biomedicines, 2022, doi:10.3390/biomedicines10112754_

Round 1

Reviewer 1 Report

Obstructive sleep apnoea (OSA) is a frequently occurring human disorder, which is associated with a number of comorbidities including hypertension, hyperlipoproteinemia and atherosclerosis. As atherosclerosis OSA is frequently paralleled by elevated levels of triglyceride-rich lipoproteins (VLDL, LDL) and impaired levels of HDL. Although the exact mechanisms of OSA-associated hyperlipoproteinemia have not been fully understood three major processes have been implicated: i) accelerated lipid synthesis in the liver, ii) impaired lipid clearance fr om the circulation, iii) activated lipolysis in the adipose tissue. These patho-mechanisms by are further complicated by accelerated oxidation of triglyceride-rich lipoproteins and marked alterations in the endocrine system. In this comprehensive review the authors summarized and critically evaluated the current knowledge on the patho-mechanisms involved in this frequently occurring but often underestimated human disorder.

In general, the authors did a good job summarizing the current status of knowledge on this frequently occurring disorder, the health political relevance of which is frequently ignored. The reference list involves more than 350 well selected citations and thus, the paper summarizes all relevant reports currently published in this field. I think this paper is of interest for a broad readership, which does not only involve expert researchers in the field but also general physicians. I am convinced that this paper will frequently be referenced and that it will improve public awareness for this disease. In principle, the paper can be published as it is but I have three suggestions for minor revision.

1. Although the paper as a whole is of high quality, chapters 7 and 8 are disappointing. They more or less involve very general statements but do not provide helpful information. In chapter 7 the authors should use their knowledge to provide detailed suggestions for further lines of research in the field. Such detailed suggestions may prompt other interested researchers to jump on the train to contribute to the advancement of knowledge in the field. Also chapter 8 consists mainly of general statements. It would certainly be helpful for the readers if the authors could specify in more detail what kind of well-designed studies they want to be carried out.

2. For this review the authors focused on human studies, which is OK with me. However, for detailed mechanistic investigations animal models are usually very helpful. It would be of interest for the readers if the authors could briefly discuss in a separate chapter whether suitable animal models are available for OSA. If so, the pros and conts of such models might be discussed. If not, would it be worth to develop such models? There are a number of mouse models that are frequently employed in atherosclerosis research. Are these models suitable for OSA research or can they be adapted for this disease?

3. In my pdf version page 14 is empty.

Author Response

Dear Reviewer,

Thank you for the valuable comments and questions. We revised the manuscript according to your suggestions and addressed all comments in a point-by-point fashion.

Q1: Although the paper as a whole is of high quality, chapters 7 and 8 are disappointing. They more or less involve very general statements but do not provide helpful information. In chapter 7 the authors should use their knowledge to provide detailed suggestions for further lines of research in the field. Such detailed suggestions may prompt other interested researchers to jump on the train to contribute to the advancement of knowledge in the field. Also chapter 8 consists mainly of general statements. It would certainly be helpful for the readers if the authors could specify in more detail what kind of well-designed studies they want to be carried out.

A1: Thank you for raising this important point. We have expanded these 2 chapters with our suggestions for the future studies. In addition, we added further clinical recommendations in other chapters supplementing chapters 7 and 8.

Q2: For this review the authors focused on human studies, which is OK with me. However, for detailed mechanistic investigations animal models are usually very helpful. It would be of interest for the readers if the authors could briefly discuss in a separate chapter whether suitable animal models are available for OSA. If so, the pros and conts of such models might be discussed. If not, would it be worth to develop such models? There are a number of mouse models that are frequently employed in atherosclerosis research. Are these models suitable for OSA research or can they be adapted for this disease?

A2: Thank you for your valuable suggestions. We have collected the main points of the animal research in this field. You can find the new chapter in Chapter 3 “ 3.1 Animal models ”. We also emphasized throughout the manuscript if the particular research was conducted in animals.

Q3: In my pdf version page 14 is empty.

A3: Thank you for your note. We have corrected the document.

We hope that in the current form our manuscript can be considered for publication in Biomedicine.

Yours sincerely,

Martina Meszaros, MD

Reviewer 2 Report

In this review the authors summarised and critically evaluated the current evidence about the possible mechanisms involved in the OSA-associated dyslipidaemia.

The introduction, figures, discussion of major findings and further research directions, concluding remarks and references are framed correctly. Overall, the article is informative for the scientific fraternity. I recommend the paper for publication after rectifying the below mentioned minor errors.

Line No.16 :    is associated with cardiovascular

Line No.40 :    in with altered

LineNo.49:      involved the OSA-associated

Line No. 66:    majority of by CM

Line No. 139:   hepatic LRP-1, CM clearance

Line No.142:   associated with elevated

Line No.205:   lecithin cholesterol acyltransferase (LCAT)

Line Nos.416/417:   LOX-1 is the main receptor for oxLDL on endothelial cells are orchestrate the    expression (line correction required)

Line No. 543:        plasma levels

Line No.634/635 : immunity is key mechanism in foam cell formation

References:

No. 137: Page No. missing

No.205: Page No. missing

No.247: Page No. missing

No.264: Page No. missing

No.348: Volume & page no. missing.

Author Response

Dear Reviewer,

Thank you for the valuable comments and questions. We revised the manuscript according to your suggestions and addressed all comments in a point-by-point fashion.

Q1: Minor errors:

Line No.16: is associated with cardiovascular

Line No.40: in with altered

Line No.49: involved the OSA-associated

Line No.66: majority of by CM

Line No.139: hepatic LRP-1, CM clearance

Line No.142: associated with elevated

Line No.205: lecithin cholesterol acyltransferase (LCAT)

Line No.416/417: LOX-1 is the main receptor for oxLDL on endothelial cells are orchestrate the    expression (line correction required)

Line No.543: plasma levels

Line No.634/635: immunity is key mechanism in foam cell formation

A1: Thank you for your valuable notes, we have corrected these typing errors. 

Q2: References:

No.137: Page No. missing

No.205: Page No. missing

No.247: Page No. missing

No.264: Page No. missing

No.348: Volume & page no. missing.

A2: Thank you for your recommendation. We have corrected the mentioned references.

We hope that in the current form our manuscript can be considered for publication in Biomedicine.

Yours sincerely,

Martina Meszaros, MD
